# Modification of Peripheral Blood Flow and Angiogenesis by CO_2_ Water-Bath Therapy in Diabetic Skeletal Muscle with or without Ischemia

**DOI:** 10.3390/biomedicines11123250

**Published:** 2023-12-08

**Authors:** Vijayan Elimban, Yan-Jun Xu, Sukhwinder K. Bhullar, Naranjan S. Dhalla

**Affiliations:** Institute of Cardiovascular Sciences, St. Boniface Hospital Albrechtsen Research Centre, Department of Physiology and Pathophysiology, Rady Faculty of Health Sciences, Max Rady College of Medicine, University of Manitoba, Winnipeg, MB R2H 2A6, Canada; velimban@sbrc.ca (V.E.); davidshu2018@gmail.com (Y.-J.X.);

**Keywords:** diabetic complications, peripheral artery disease, CO_2_ water-bath therapy, skeletal muscle angiogenesis

## Abstract

Previously, it was shown that both blood flow and angiogenesis in the ischemic hind limb of diabetic rats were increased upon CO_2_ treatment for 4 weeks. In the present study, we have compared the effects of 6 weeks CO_2_ therapy in diabetic rats with or without peripheral ischemia. Diabetes was induced in rats by a tail vein injection of streptozotocin (65 mg/kg body weight), whereas peripheral ischemia was produced by occluding the femoral artery at 2 weeks of inducing diabetes. Both diabetic and diabetic-ischemic animals were treated with or without CO_2_ water-bath at 37 °C for 6 weeks (30 min/day; 5 days/week) starting at 2 weeks, after the induction of ischemia. CO_2_ treatment did not affect heart rate and R-R interval as well as plasma levels of creatine kinase, glucose, cholesterol, triglycerides and high density lipoproteins. Unlike the levels of plasma Ox-LDL, MDA and TNF-α, the levels of NO in diabetic group were increased by CO_2_ water-bath treatment. On the other hand, the levels of plasma Ox-LDL and MDA were decreased whereas that of NO was increased without any changes in TNF-α level in diabetic-ischemic animals upon CO_2_ therapy. Treatment of diabetic animals with CO_2_ increased peak, mean and minimal blood flow by 20, 49 and 43% whereas these values were increased by 53, 26 and 80% in the diabetic-ischemic group by CO_2_ therapy, respectively. Blood vessel count in diabetic and diabetic-ischemic skeletal muscles was increased by 73 and 136% by CO_2_ therapy, respectively. These data indicate that peripheral ischemia augmented the increase in blood flow and development of angiogenesis in diabetic skeletal muscle upon CO_2_ therapy. It is suggested that greater beneficial effects of CO_2_ therapy in diabetic-ischemic animals in comparison to diabetic group may be a consequence of difference of changes in the redox-sensitive signal transduction mechanisms.

## 1. Introduction

Diabetes is a chronic metabolic disorder manifested by hyperglycemia and if left untreated or improperly managed, it may result in complications such as heart disease [1]. Diabetes is also known to produce atherosclerosis leading to occlusive disease of the hind limbs, which is commonly referred to as peripheral artery disease (PAD) [2]. The advanced stage of PAD is characterized by chronic limb-threatening ischemia with very high morbidity and mortality rates [3,4,5,6]. It has also been shown that diabetes is a significant PAD risk factor in addition to hypertension, chronic kidney disease, hyperlipidemia, and smoking [7]. PAD begins early in diabetic patients and progresses rapidly but remains mostly asymptomatic, which makes it difficult to detect. However, strict management conditions for PAD in diabetes have been reported to promote early diagnosis and reduce progression [7]. Although several medical therapies such as antidiabetic, antiatherosclerotic, antithrombotic, antihypertensive and vasodilatory drugs as well as surgical procedures are available for the treatment of PAD [7,8,9,10,11,12,13], none of these interventions are entirely satisfactory. It is also pointed out that more than 200 million patients all over the world are affected by PAD [14] and thus there is an urgent need to find solution for this health hazard.

Bathing in naturally enriched carbon dioxide (CO_2_) spring waters known as balneotherapy, has been used as a traditional method, for over a century, for health promotion and prevention of PAD as well as treatment and rehabilitation of patients suffering from various diseases [15,16,17,18]. Such beneficial effects of balneotherapy are considered to be due to marked vasodilatory and anti-inflammatory actions of CO_2_. In fact, both clinical and experimental investigations have revealed that CO_2_ therapy improves blood flow in the ischemic limb in PAD [4,19,20,21,22,23,24,25,26,27,28]. Due to the microcirculation promoting effect of CO_2_, angiography using CO_2_ has been employed to monitor immediate response to endovascular therapy of PAD [29,30,31,32]. Since CO_2_ water-bathing induces marked changes in body surface temperature, thermal imaging has also been used to measure the actions of CO_2_ treatment in augmenting blood flow in patients with PAD [33]. Thus, CO_2_ has not only been used as a therapeutic intervention for promoting blood flow but has also been exploited as a diagnostic tool for the detection and progression of PAD.

In view of higher incidence of ulcers in patients with diabetes, bathing in CO_2_-enriched water has been reported to accelerate healing of foot ulcers [28]. We have also observed that 4 weeks of CO_2_ therapy augmented the blood flow and angiogenesis in the skeletal muscle of 6 weeks diabetic-ischemic hind limb of rats [25]. Since the effects of CO_2_ water-bath therapy on blood flow as well as angiogenesis in diabetic animal hind limb without ischemia have not been examined, this study was undertaken to compare the effectiveness of CO_2_ water-bath therapy in diabetic and diabetic-ischemic hind limb in promoting blood flow and angiogenesis. Diabetic animals with or without peripheral ischemia also received water-bath treatment in the absence or presence of CO_2_ to examine changes in some biomarkers for inflammation and oxidative stress.

## 2. Materials and Methods

### 2.1. Induction of Diabetes and Hind Limb Ischemia

Diabetes was induced in male Sprague-Dawley rats (175–200 g) by a tail vein injection of streptozotocin (65 mg/kg body mass) according to the procedure used in our laboratory [34,35]. The animals were divided into 6 groups as shown in Figure 1. Two weeks after the induction of diabetes, 3 groups of animals were anesthetized with 1–5% isoflurane in oxygen at a flow rate of 2 L/min, for the induction of hind limb ischemia according to the method described earlier [24,25]. Inguinal artery in the left thigh was occluded by ligation using 3-0 surgical silk [36]. The wound was closed and the animals were allowed to recover in their cages. Animals with no infection at the surgical site and the wounds completely healed were used for experiments.

### 2.2. CO_2_-Enriched Water-Bath Therapy

Two weeks after the hind limb ischemia or four weeks after inducing diabetes, animals were subjected to no water-bath or water-bath treatment at 37 °C for 30 min daily (5 days per week) for 6 weeks as described in Figure 1 [25]. Two groups were subjected to CO_2_ water-bath (diabetic and diabetic-ischemic with CO_2_ mixing), whereas other two groups were exposed to water-bath without CO_2_ mixing (diabetic and diabetic-ischemic without CO_2_ mixing). Another two groups were not exposed to any water-bath treatment (diabetic and diabetic-ischemic). Although other investigators have used four weeks for CO_2_ treatment [21,23], in this study we used 6 weeks treatment in order to assess if prolonged treatment results in improved effects in comparison to our earlier 4 weeks treatment study [25]. Carbothera, a therapeutic footbath unit (Mitsubishi Rayon Engineering, Tokyo, Japan) containing a multi-layered composite hollow-fiber membrane, was used for dissolving CO_2_ in tap water [37]. The CO_2_ concentration was 1000–12,000 ppm (pH 4.5–4.6) and temperature of the bath was kept constant at 37 °C. The animals were kept in a Plexiglas plastic rat restrainer with several large perforations underneath, and the hind limb was immersed in the water-bath. At the end of 6 weeks treatment, the animals were anesthetized with ketamine/xylazine (90/9 mg/kg body weight). Blood was collected from the abdominal aorta, and serum was used for biochemical analysis. Left leg skeletal muscle was also dissected at the ischemic site and stored at −70 °C for analysis.

### 2.3. Measurements of Blood Flow

Blood flow to the hind limb was measured after femoral artery ligation as well as at 6 weeks after CO_2_ treatment using a Pulse Wave Doppler System (Indus Instruments, Webster, TX, USA) [38,39]. Anesthetized rats were tied in a supine position on the ECG board (THM100, Indus Instruments). Blood flow was measured using a 20-MHz pulsed Doppler probe. Data analysis was carried out with an Indus Instruments Doppler Signal Processing Workstation whereas Doppler spectrogram analysis software was used for calculating the heart rate and R-R interval. The methods were same as described previously [24,25].

### 2.4. Assessment of Angiogenesis

Skeletal muscle tissue above the femoral artery ligation was fixed with 10% formalin for 24–48 h and embedded in paraffin [24]. About 10–15 sections of each sample at 5 µm were taken using a Shandon, Finesse 325 microtome. After staining with Hematoxylin and Eosin, all tissue sections were examined under light microscope. Blood vessels were visualized using Image ProPlus digital system Version 7.0 [40]. The analysis of angiogenesis was carried out by a technician blinded to the animal protocol. It is mentioned that the data are expressed per tissue section, although the area examined in all groups was not the same.

### 2.5. Serum Analysis

Serum analysis was carried out with the Roche Cobas 6000 module c501 (Roche Diagnostics GmbH, Mannheim, Germany) automated system as suggested earlier [25]. Enzymatic–colorimetric assays were used for the measurement of cholesterol, triglycerides, and high density lipoproteins (HDLs). Glucose was measured with a hexokinase UV method whereas creatine kinase (CK) was estimated using N-acetyl cysteine-activated UV assay. Serum tumor necrosis factor α (TNF-α) and NO (nitric oxide) were measured using methods developed by R&D Systems, Minneapolis, MN, USA and Enzo Life Sciences, Farmingdale, NY, USA, respectively. Malondialdehyde (MDA) as well as oxidized low-density lipoproteins (Ox-LDLs) (Zeptometrix Corporation, Buffalo, NY, USA) were estimated using specific ELISA kits according to procedure by the manufacturer.

### 2.6. Data Analysis

Each value is expressed as mean ± standard error (SE). The statistical analysis was carried out using Origin Version 6 software (Microcal Software Inc., Northampton, MA, USA). The results from the same group were analyzed using one-way analysis of variance (ANOVA) whereas those from different groups were analyzed using Student’s *t* test and *p* value < 0.05 was considered significant.

## 3. Results

### 3.1. Characteristics of Animal Models with or without CO_2_ Therapy

The data in Table 1 show the values for body weight, heart rate, R-R interval and plasma creatine kinase levels in both 4 weeks diabetic and diabetic-ischemia rats subjected to no water-bath, water-bath without CO_2_ and CO_2_ water-bath treatments for a period of 6 weeks. There was no difference in these parameters with or without the water-bath treatments for both diabetic as well as diabetic-ischemic animals. Furthermore, the values for plasma glucose, cholesterol, triglycerides and high density lipoproteins in both diabetic and diabetic-ischemic animals were not affected by treatments with water-bath with or without CO_2_ mixing. These results did not show any difference between diabetic and diabetic-ischemic animals with respect to cardiac function and metabolic status (Table 2).

In order to test the effects of peripheral ischemia in diabetic animals, peak blood flow, mean blood flow and minimal blood flow were measured in diabetic and diabetic-ischemic animals without any treatment with water-bath. Peak, mean and minimal blood flow in diabetic-ischemic animals were 16, 55 and 34% lower than those in the diabetic animals (Figure 2). However, the vessel counts in the hind limb skeletal muscle of the diabetic and diabetic-ischemic groups were not different from each other (Figure 2). These results show that the effects of peripheral ischemia for depression in the blood flow in diabetic rats were not associated with any changes in the density of small blood vessels in the skeletal muscle.

### 3.2. Influence of CO_2_ Therapy on Blood Flow

Figure 3 shows the effect of CO_2_ therapy on blood flow in the hind limb of both diabetic and diabetic-ischemic rats. The values for peak, mean and minimal blood flow were 20, 40 and 43% higher in diabetic animals upon treatment with CO_2_ water-bath in comparison to those with water-bath without CO_2_ mixing. On the other hand, the values for peak, mean, and minimal blood flow were 53, 26 and 80% higher in diabetic-ischemic group upon CO_2_ therapy in comparison to those with water-bath without CO_2_ mixing. These results show that increase in both peak and minimal blood flow values in diabetic-ischemic animals due to CO_2_ therapy were greater than those observed in the diabetic group. Although the increase in the mean blood flow in the diabetic-ischemic group due to CO_2_ treatment was not higher than that in the diabetic animals, such results may be due to a relatively high increase in minimal blood flow values in these animals.

### 3.3. Influence of CO_2_ Therapy on Angiogenesis in Skeletal Muscle

CO_2_ treatment of diabetic animals was observed to decrease the blood vessel count by about 27% in comparison to that seen in the skeletal muscle upon treatment with water-bath with no CO_2_ mixing. On the other hand, blood vessel count in the diabetic-ischemic group was increased by about 136% due to CO_2_ treatment. These results (Figure 4) indicate that the development of angiogenesis in diabetic-ischemic animals was markedly augmented by CO_2_ therapy.

### 3.4. Effects of CO_2_ Therapy on Plasma Biomarkers

It was interesting to observe that the values of plasma Ox-LDL, MDA, NO and TNF-α in diabetic animals upon water-bath without CO_2_ mixing were significantly lower than those without water-bath treatment (Figure 5). On the other hand, the values of plasma Ox-LDL and TNF-α were decreased whereas those for plasma MDA and NO were increased in the diabetic-ischemic group upon water-bath without CO_2_ treatment (Figure 5). Although the exact reason for this differential effect of water-bath without CO_2_ treatment in diabetic and diabetic-ischemic groups with respect to these biomarkers is not clear at present, the involvement of some stress-induced changes cannot be ruled out. It can be argued that Ox-LDL/LDL and MDA/LDL ratios are more sensitive biomarkers of oxidative stress than Ox-LDL and MDA levels per se but we did not measure plasma LDL levels in this study and thus values for these indices are not given.

It should be noted that therapy of diabetic animals with CO_2_ was observed to increase the plasma levels of NO, unlike plasma levels for Ox-LDL, MDA and TNF-α. Furthermore, plasma levels of Ox-LDL and MDA were significantly decreased whereas that for plasma NO was increased without any changes in TNF-α level in the diabetic-ischemic group with CO_2_ therapy. These results indicate that CO_2_ therapy improved plasma level of NO in diabetic animals whereas in diabetic-ischemic group plasma NO level was further increased and plasma levels of Ox-LDL and MDA were significantly depressed.

## 4. Discussion

In this study, peak, mean and minimal blood flow in the hind limb of diabetic rats were increased by 20, 49 and 43% due to CO_2_ therapy for a period of 6 weeks. On the other hand, these parameters in the hind leg of normal rats were increased by CO_2_ therapy by 2 to 4 folds under similar conditions [24,26]. Attenuated responses of diabetic animals to CO_2_ water-bath therapy in comparison to normal rats may be due to differences in the modification of both extrinsic and intrinsic factors, which are known to regulate the peripheral blood flow [4,6,11]. In this regard, it is pointed out that diabetes has been shown to induce marked alterations in Ca^2+^- handling as well as metabolism in the skeletal muscle [34,35,41]. Although changes in the sympathetic activity due to CO_2_ therapy have been suggested to explain the improved blood flow [42], this mechanism cannot account for the observed increase in blood flow because no difference for various indices, such as heart rate, R-R interval and plasma creatine kinase levels in diabetic animals, was seen between water-bath treatments with or without CO_2_ mixing. Furthermore, alterations in blood viscosity cannot be considered to explain the difference in blood flow [4], because the levels of plasma cholesterol, triglycerides, high density proteins and glucose in the diabetic animals with water-bath treatments in absence or presence of CO_2_ mixing were similar. Although development of angiogenesis by CO_2_ therapy has been reported to result in promoting the blood flow in skeletal muscle [24], this may not be the case in this condition because the blood vessel count in diabetic skeletal muscle was decreased upon CO_2_ therapy. Since plasma levels of NO, a well known vasodilator, were increased by CO_2_ treatment of diabetic animals, it appears that the observed increase in blood flow under this condition may be a consequence of increase in the plasma level of NO.

In contrast to diabetic animals, diabetic-ischemic group showed greater increase in peak, mean and minimal blood flow (53, 26 and 80%, respectively) and a marked increase (136%) in the skeletal muscle blood vessel count upon CO_2_ treatment. These observations support the view that peripheral ischemia augments the increase in blood flow as well as the development of angiogenesis due to CO_2_ therapy and are in agreement with our early report [25]. Such beneficial effects of CO_2_ therapy in diabetic-ischemic group were associated with significant depressions in the level of plasma Ox-LDL and MDA levels as well as further increase in the level of plasma NO. The observed alterations in plasma biomarkers indicate that the effects of CO_2_ may be related to reduction in the degree of oxidative stress in the diabetic-ischemic skeletal muscle as well as direct action of NO on the peripheral vasculature. In this regard, it is pointed out that several studies have shown that CO_2_ is an antioxidant gaso-transmitter [43,44,45] and NO is involved in promoting angiogenesis in the different organs including hind limb [36,46,47]. Furthermore, CO_2_ has been demonstrated to increase cyclic nucleotides and prostanoids for vascular regulation [48]. Thus, it is likely that the modification of both blood flow and angiogenesis in the hind limb of diabetic-ischemic animals by CO_2_ therapy may be mediated through different mechanisms including redox-sensitive signal transduction pathway.

Although in this experimental study we have shown improvement in peripheral blood flow in diabetic-ischemic animals upon treatment with CO_2_ water-bath at 37 °C, it is pointed out that such a beneficial effect of this mode of therapy in normal-ischemic animals has been demonstrated to be dependent upon temperature of the water-bath [26]. Furthermore, in view of our observations in this study as well as previous investigation [25], it appears that the effectiveness of CO_2_ water-bath therapy is influenced by the duration of diabetes. Thus, extensive work needs to be carried out to optimize this technique by employing the diabetic-ischemic animals before this therapy can be considered for clinical application. Since diabetes is associated with several risk factors including hypertension for inducing vascular complications, it would be valuable to examine the effectiveness of CO_2_ water-bath therapy by employing diabetic-hypertensive animals. Nonetheless, a preliminary study in diabetic patients with foot ulcers has shown some promising results with CO_2_ water-bath treatment [28], and it is hoped that a large clinical trial will be undertaken to establish the beneficial effects of CO_2_ therapy for the treatment of vascular complications in diabetic patients.

## 5. Conclusions

This study has demonstrated that CO_2_ water-bath therapy improves blood flow and increase plasma level of NO in diabetic rats. Furthermore, peripheral ischemia in diabetic animals was observed to augment the effects of CO_2_ treatment on blood flow and induce angiogenesis in the hind limb. It is proposed that the beneficial effects of CO_2_ therapy on peripheral blood flow and angiogenesis in diabetic-ischemic animals may be elicited through the involvement of redox-sensitive and NO-related signal transduction pathways.

## Figures and Tables

**Figure 1 biomedicines-11-03250-f001:**
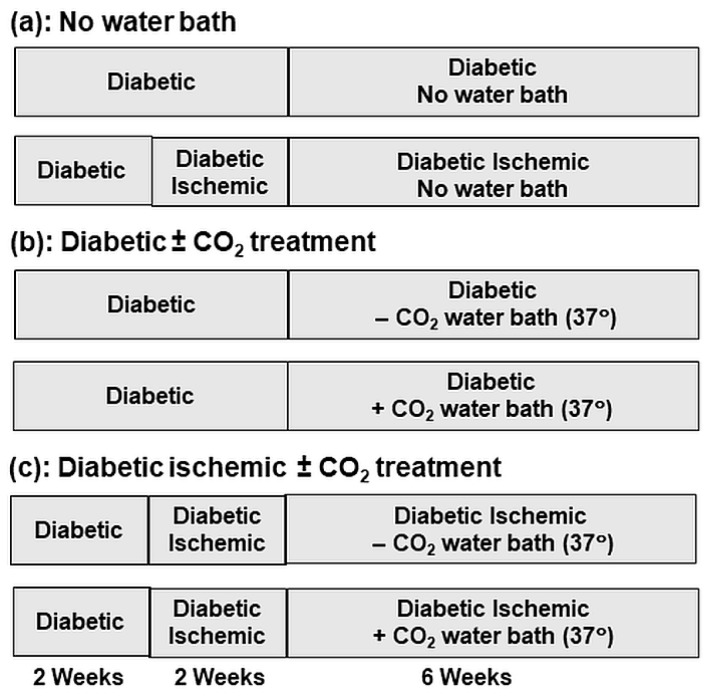
Diabetic rat groups showing time periods for inducing ischemia in diabetic rats followed by treatment with (+) or without (−) CO_2_ water-bath for 6 weeks.

**Figure 2 biomedicines-11-03250-f002:**
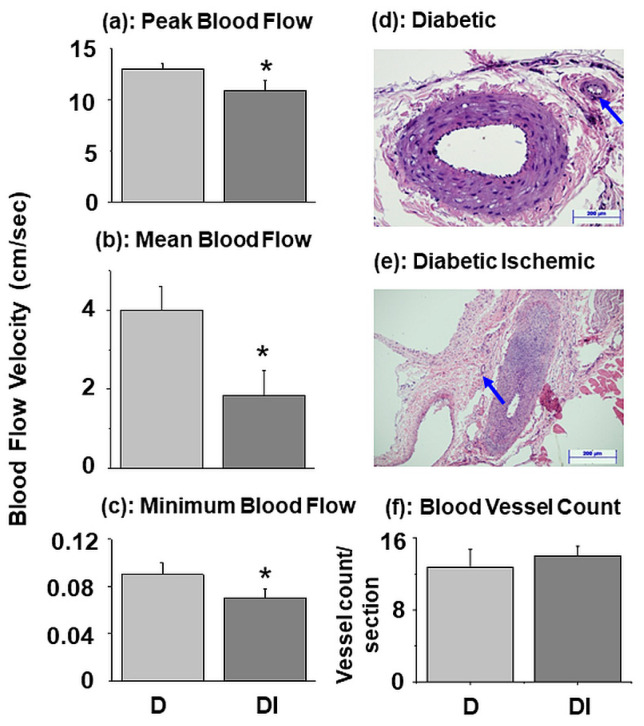
Peak (**a**), mean (**b**), and minimal (**c**) blood flow in hind limb skeletal muscle in 10-week diabetic as well as diabetic-ischemic rats without any water bath treatment. Values are means ± SE of 5 animals per group. *—*p* < 0.05 vs. respective diabetic group. Representative hind limb skeletal muscle sections of diabetic (**d**), diabetic-ischemic (**e**) and their blood vessel count (**f**) are also shown. Scale bars = 200 µm. Arrows indicate the location of blood vessel.

**Figure 3 biomedicines-11-03250-f003:**
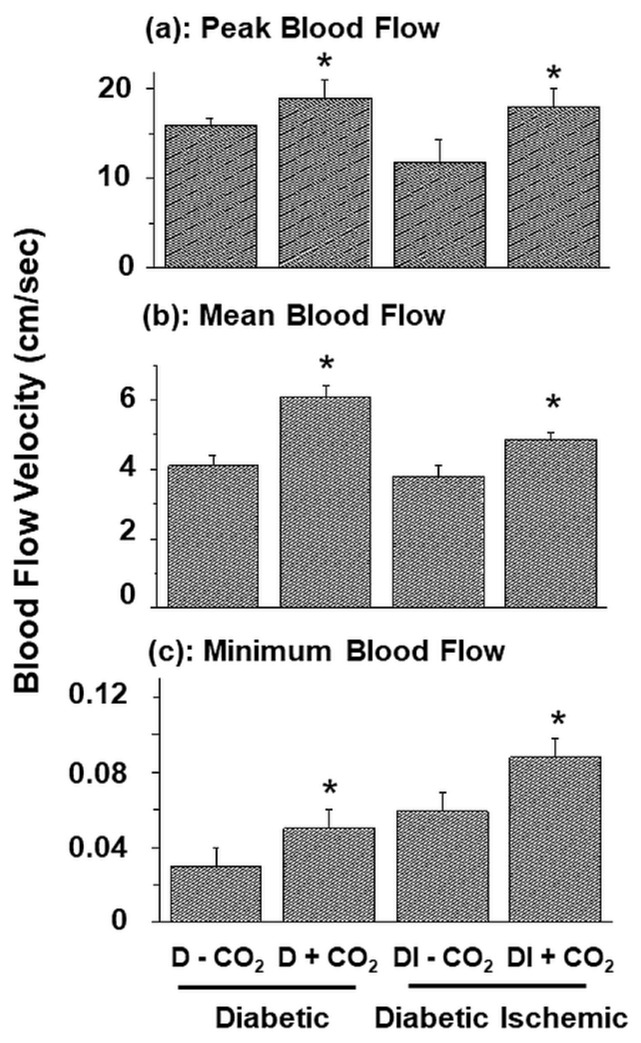
Influence of 6-week CO_2_ water-bath treatment on the peak (**a**), mean (**b**), and minimal (**c**) blood flow in hind limb skeletal muscle in 2-week diabetic rats with or without 2 weeks of peripheral ischemia (I). Values are mean ± SE of 5 to 6 animals per group. *—*p* < 0.05 vs. respective group without CO_2_.

**Figure 4 biomedicines-11-03250-f004:**
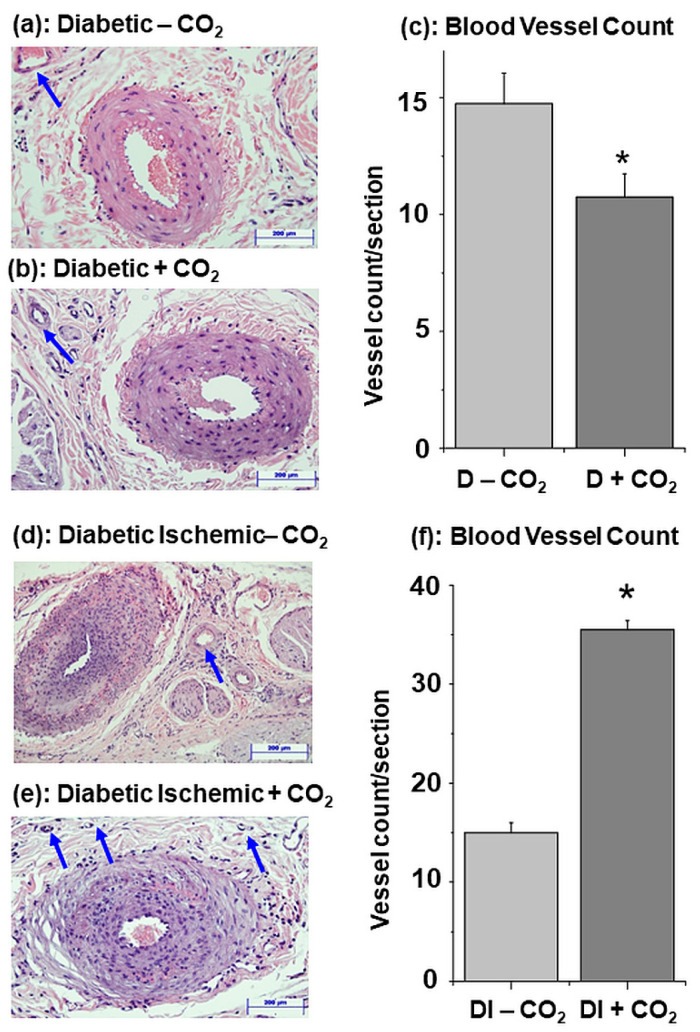
Representative hind limb skeletal muscle sections of diabetic without CO_2_ (**a**), diabetic with CO_2_ (**b**), diabetic-ischemic without CO_2_ (**d**), and diabetic-ischemic with CO_2_ (**e**) water-bath therapy at 37 °C. Bar graphs in (**c**,**f**) show their respective blood vessel count. Values are means ± SE of 6 animals per group. *—*p* < 0.05 vs. respective group without CO_2_. Arrows indicate capillaries and small vessels in the tissue sections. Scale bars = 200 µm.

**Figure 5 biomedicines-11-03250-f005:**
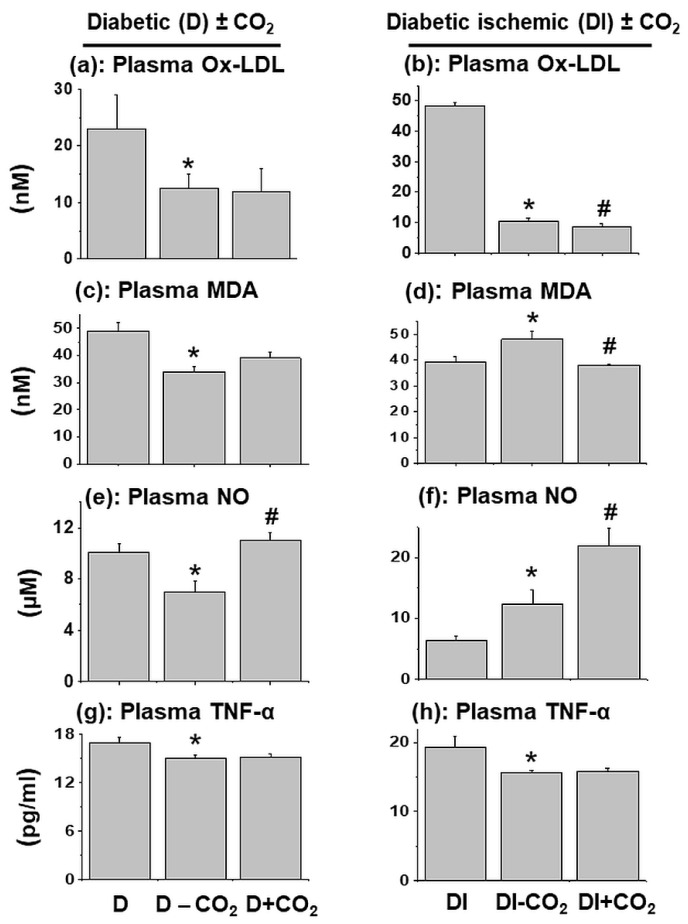
Plasma oxidized lipoproteins (Ox-LDLs) (**a**,**b**), malondialdehyde (MDA) (**c**,**d**), nitric oxide (NO) (**e**,**f**), and tumor necrotic factor—α (TNF-α) (**g**,**h**) in diabetic and diabetic-ischemic rats with or without CO_2_ treatment, respectively. Values are means ± SE of 5 to 6 animals per group. *—*p* < 0.05 vs. respective control diabetic group. #—*p* < 0.05 vs. ischemia-diabetic group.

**Table 1 biomedicines-11-03250-t001:** Effect of CO_2_ treatment for 6 weeks at 37 °C on body weight, heart rate, R-R interval and plasma creatine kinase levels in 4 weeks diabetic and diabetic- ischemic rats.

Groups	Body Weight (g)	Heart Rate (beats/min)	R-R Interval (ms)	Creatine Kinase (U/L)
**(a): Diabetic rats**				
No water bath	407 ± 13	286 ± 12	211 ± 9	387 ± 36
Water bath − CO_2_	443 ± 11	295 ± 14	211 ± 9	397 ± 69
Water bath + CO_2_	426 ± 22	299 ± 15	205 ± 5	492 ± 77
**(b): Diabetic-ischemic rats**				
No water bath	383 ± 28	286 ± 13	213 ± 13	482 ± 92
Water bath − CO_2_	382 ± 21	292 ± 16	208 ± 11	398 ± 70
Water bath + CO_2_	422 ± 14	284 ± 8	213 ± 6	496 ± 66

Values are mean ± SE of 6 animals in each group. Values for heart rate, R-R interval and plasma creatine kinase levels in diabetic animals were not significantly (*p* > 0.05) different from those for the control rats.

**Table 2 biomedicines-11-03250-t002:** Effect of CO_2_ treatment for 6 weeks at 37 °C on plasma glucose and lipid levels in 4 weeks diabetic and diabetic-ischemic rats.

Groups	Glucose (mM)	Cholesterol (mM)	Triglycerides (mM)	High Density Lipoproteins (U/L)
**(a): Diabetic rats**				
No water bath	34.2 ± 1.4	2.88 ± 0.31	9.51 ± 4.99	0.91 ± 0.20
Water bath − CO_2_	35.0 ± 1.2	2.28 ± 0.37	5.99 ±1.78	1.28 ± 0.11
Water bath + CO_2_	34.0 ± 0.2	2.98 ± 1.06	9.55 ± 4.11	1.30 ± 0.24
**(b): Diabetic-ischemic rats**				
No water bath	37.0 ± 1.9	2.78 ± 0.34	5.35 ± 3.84	1.60 ± 0.23
Water bath − CO_2_	38.8 ± 1.1	2.78 ± 0.44	6.34 ± 3.50	0.98 ± 0.19
Water bath + CO_2_	36.0 ± 1.9	2.10 ± 0.14	5.27 ± 1.94	1.30 ± 0.10

Values are mean ± SE of 5 to 6 animals in each group. Plasma levels of glucose, cholesterol and triglycerides in diabetic animals were about 100%, 80% and 50% higher than those in control rats, respectively, without any difference in the levels for high density lipoproteins.

## Data Availability

Data are contained within the article.

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
