# Peer review of "Modification of Peripheral Blood Flow and Angiogenesis by CO2 Water-Bath Therapy in Diabetic Skeletal Muscle with or without Ischemia"

_biomedicines, 2023, doi:10.3390/biomedicines11123250_

Round 1
Reviewer 1 Report
Comments and Suggestions for Authors
Thank you for inviting me to review this article addressing the peripheral arterial ischemia, a pathology often underdiagnosed and undertreated adequately in clinical practice. Although this paper it is addressed to an experimental study, I think that this approach has the potential to be further applied in practice. Vascular complications associated with diabetic foot have multiple prognostic implications in addition to limiting functional status and leading to decreased quality of life for these patients. Moreover, diabetic foot may be a marker of cardiovascular risk, which is associated with an increased risk of an acute vascular event.
Thus, a deeper understanding of the underlying mechanisms and finding available and reproducible methods to follow the local evolution in order to prevent the complications, it is very useful and beneficial. From this point of view, I think that the manuscript is original and of scientific interest and sustains a larger use of this method in practice. The current study has a clear objective and design, the methodology is well described, the results are clearly illustrated by tables and figures. I especially appreciate the section of discussion which is well organized and it is sustained by updated data as reflected in the selected references.
On the other hand, we have to take into account that the results of this approach are considerably influenced by the duration of diabetes evolution, association of other frequent risk factors like smoking and hypertension, the medication and adherence to therapy. In this respect, I encourage the authors to extend the discussion about the limits of their study and also to investigate the beneficial role of this method in animals depending on the age of diabetes and the presence of other associated risk factors.
Author Response
As suggested we have now added one paragraph at the end of Discussion section (#4: lines 286 to 300) to indicate that further studies are needed for optimizing the CO2 water-bath therapy in diabetic-ischemic animals before considering its clinical application. Since vascular complications in diabetes are influenced by different risk factors including hypertension, it is suggested that diabetic-hypertensive animals be also employed for investigating the effectiveness of CO2 water-bath therapy. Furthermore, a large clinical trial in diabetic patients is warranted to establish the therapeutic potential of this intervention.
Reviewer 2 Report
Comments and Suggestions for Authors
The authors investigated the effects of CO2 water-bath therapy in diabetic rats with and without peripheral ischemia. However, I have one minor comment.
1) The authors described the changes in plasma ox-LDL and MDA levels with or without CO2 water-bath therapy. However, please show the changes of the ratios of ox-LDL/LDL or MDA/LDL.
Author Response
Since we did not measure changes in the CO2 levels in this study, we are unable to show the values for Ox LDL/LDL or MDA/LDL ratios. However, the significance of these indices has now been added in the result section (#3.4- line 229 to 232).